# The Cardiometabolic Health Benefits of Sauna Exposure in Individuals with High-Stress Occupations. A Mechanistic Review

**DOI:** 10.3390/ijerph18031105

**Published:** 2021-01-27

**Authors:** Kaemmer N. Henderson, Lauren G. Killen, Eric K. O’Neal, Hunter S. Waldman

**Affiliations:** Human Performance Laboratory, Department of Kinesiology, University of North Alabama, Florence, AL 35632, USA; khenderson2@una.edu (K.N.H.); lkillen1@una.edu (L.G.K.); eoneal1@una.edu (E.K.O.)

**Keywords:** firefighters, police, military, oxidative stress, inflammation, performance

## Abstract

Components of the metabolic syndrome (i.e., hypertension, insulin resistance, obesity, atherosclerosis) are a leading cause of death in the United States and result in low-grade chronic inflammation, excessive oxidative stress, and the eventual development of cardiometabolic diseases (CMD). High-stress occupations (HSO: firefighters, police, military personnel, first responders, etc.) increase the risk of developing CMD because they expose individuals to chronic and multiple stressors (i.e., sleep deprivation, poor nutrition habits, lack of physical activity, psychological stress). Interestingly, heat exposure and, more specifically, sauna bathing have been shown to improve multiple markers of CMD, potentially acting as hormetic stressors, at the cellular level and in the whole organism. Therefore, sauna bathing might be a practical and alternative intervention for disease prevention for individuals with HSO. The purpose of this review is to detail the mechanisms and pathways involved in the response to both acute and chronic sauna bathing and collectively present sauna bathing as a potential treatment, in addition to current standard of care, for mitigating CMD to both clinicians and individuals serving in HSO.

## 1. Introduction

Cardiovascular disease is the leading cause of death in the United States among both men and women, accounting for one in every four deaths annually [1]. Moreover, high-stress occupations (HSO: military personnel, firefighters, police officers, etc.) have an elevated risk for the development of cardiometabolic diseases (CMD) such as atherosclerosis [2], obesity [3], heart disease [4], and sudden cardiac death [3], due to the multiple stressors encountered within their occupation. Additionally, chronic exposure to stressors such as heat and smoke [5,6], elevated inflammation and oxidative stress (OS) levels [7,8], intense physical exertion [5,6,9], psychological stress [6,10], nutrient-poor diets [11], and irregular sleep patterns [12] can further elevate the risk for developing CMD.

Recent research conducted by Hunter et al. demonstrated that in firefighters undergoing a fire simulation training consisting of a victim search and rescue while suppressing a live fire, thrombus production increased by 73%, and platelet-monocyte binding by 7%, resulting in impaired vascular functioning after the subject’s physiological values returned to baseline [13]. Additionally, there was a rise in fibrinolytic capacity and asymptomatic myocardial ischemia and a significant increase (~1.6 ng/L; *p* < 0.01) in cardiac troponin I concentration after exposure to the live-fire drill [13]. It is not surprising that the leading cause of on-duty fatalities (~45%) among firefighters are attributed to CMD and sudden cardiac deaths [4,14]. Data on policing collected across a 10-year (2000–2010) period demonstrated that approximately 7% of law enforcement deaths were due to myocardial infarctions [15], and subsequent investigations have found that law enforcement personnel have an increased risk for cardiovascular morbidity compared to the general population [16,17]. McAllister et al. showed that an active shooter drill lasting ~50 s elevated epinephrine, α-amylase, and secretory immunoglobin A, which are known blood markers of increased stress and inflammation [18]. Finally, observational data reporting on the health of personnel in the United States Army found that active duty soldiers exhibit higher prevalence rates of CMD compared to all other chronic diseases (e.g., cancer, dementia, etc.) [19]. In fact, over half (~51%) of military service members are considered too overweight or obese (body mass index >25.0 kg/m^2^) to perform their military-specific duties [20]. Collectively, these data demonstrate that individuals serving in HSO work in environments that promote physiological and psychological stress and exacerbate markers of CMD. Although implementing an exercise [21] or nutrition intervention [22,23] could result in positive outcomes, due to poor adherence to nutrition [24,25] and exercise programs [26], alternative and practical interventions are needed for mitigating CMD in HSO.

One alternative option for potentially improving cardiometabolic health in HSO is sauna bathing. There are two types of saunas: wet sauna and dry sauna. Wet saunas (temperature = 70–100 °C; humidity ≥ 50%) are specifically designed to increase the thermal load of an individual by maintaining a high internal humidity to reduce evaporative cooling [27], whereas dry saunas (temperature = 80–90 °C; humidity = 10–20%) are typically made of wood and heated by an electrical heater and are the subject of most clinical research [28]. Acute (≤3 days) and chronic (>1 week) sauna bathing has shown various health benefits such as improving respiratory function [29], markers of cardiovascular health [30,31], and lipid profiles [32], as well as increasing overall longevity for chronic users. According to Laukkanen et al., on the basis of the Kuopio Ischemic Heart Risk Factor, increased duration and frequency of sauna bathing decreased the risk of fatal cardiovascular disease incidences [31]. After a 15-year follow-up, a dose–response relationship was observed among sauna frequency and cardiovascular mortality rates. Participants who attended one, two to three, or greater than four sauna session a week had a reduced cardiovascular mortality rate of 10.1, 7.6, and 2.7 per 1000 person-years, respectively [31].

Provided that healthcare spending will likely exceed three trillion this year and that ~75% of those funds will be directed towards modifiable risk factors such as heart disease, diabetes, obesity, and hypertension, clinicians and researchers alike are garnering interest in proactive and preventive interventions rather than reactive approaches. In the present review, several types of heat stress studies (i.e., Waon therapy, Japanese *onsen* bathing, localized heating, etc.) were examined to draw implications and prescribe practical guidelines for individuals with an HSO. However, considering that the bulk of research incorporating human models has implemented dry saunas likely due to the fact that researchers are interested in the health effects and physiological responses of direct-passive heat as opposed to moist heat, this narrative review will mainly focus on the cellular and whole-body mechanisms that dry sauna bathing might modulate and that could improve the cardiometabolic profile of these men and women.

## 2. An Overview of Stressors and High-Stress Occupations

Due to the nature of their work, individuals with an HSO are chronically exposed to various stressors. For the purpose of this review, stress is defined as an external event or condition that places a strain on a biological system [33]. Stress has two components: an acute phase and a chronic phase. In an acute setting (~2 min–3 days), stress is generally beneficial, as acute exposure to reactive oxygen species and moderate OS can upregulate mitochondrial adaptations such as mitochondrial fusion and mitochondrial biogenesis, resulting in a more robust intracellular antioxidant system. Additionally, the acute activation of the autonomic nervous system increases the release of glucocorticoids and catecholamines, effectively altering an organism’s metabolic rate and increasing the activity of transcription factors [34]. Over time, repeated exposure to acute stressors will result in an overall stronger organism, as a new physiological state or equilibrium is achieved through a process known as heterostasis [35]. However, chronic stress (>3 days) stimulates the endocrine system and can result in excessive activation of the hypothalamic pituitary adrenal axes and sympathoadrenal axes, leading to a dysregulated redox system and excessive OS, epinephrine, and cortisol secretion [36,37]. Rozanski et al. verified a direct association between chronic stress and CMD [38], as chronic stress is known to modulate vascular endothelial cell function and platelet aggregation [38].

Regarding HSO, a unique aspect for individuals serving in these occupations are the simultaneous exposure to both psychological and physiological stressors (i.e., dual stressors). Dual stressors are known to elicit greater activation of the hypothalamic and sympathoadrenal axes and subsequent greater release of stress markers (cortisol, epinephrine, norepinephrine), compared to a single stressor [36,39]. Increased levels of cortisol and OS in the body can upregulate several pro-inflammatory pathways which result in the development of several cardiometabolic and neurological diseases, such as atherosclerosis, diabetes, rheumatoid arthritis, and dementia [40,41]. Due to the lack of adherence to exercise and nutrition interventions [25,26] and the time restraints forced on an individual working in an HSO (i.e., 24-h work cycles and/or shift work), practical interventions such as sauna bathing warrant additional attention.

## 3. Physiological Responses to Acute Heat Exposure

Sauna bathing and heat exposure are acute stressors that invoke a plethora of physiological responses. Indeed, it would seem counterintuitive to expose a highly stressed population to an additional stressor; however, sauna bathing has emerged in the literature as a potential intervention for improving multiple markers of CMD, and the present data demonstrate a trend of positive physiological effects [32,42,43,44,45]. Although data are scarce, a recent review of endocrine responses to sauna bathing showed that some markers of stress (i.e., cortisol, β-endorphins, and adrenocorticotropic hormone) respond in a highly variable manner to acute heat exposure [46]. These differences in hormone response are likely due to methodology differences such as sauna bathing duration, temperature, and weekly frequency visits considered in each study. Therefore, until additional research is conducted in this area, expanding upon cardiovascular responses will likely provide the reader a more comprehensive view of the physiological responses to sauna bathing.

Following a single session, sauna bathing has been shown to acutely decrease blood pressure compared to pre-sauna measures (*p* < 0.05) [47,48] and improve markers of cardiovascular function, such as arterial stiffness [48]. Acute exposure to heat produces a state of mild hyperthermia and an elevation in core body temperature (>37 °C), resulting in the redistribution of blood towards the skin to promote sweating [49]. The removal of some metabolic waste by-products (e.g., carbon dioxide and sodium) as well as of water is increased [44] and causes a rise in ventilation, blood pressure, heart rate, and systemic cardiac output to compensate for the body’s increased need of oxygen [50]. Moreover, acute sauna bathing (~30 min) has been shown to improve the endothelial function via nitric oxide release and a subsequent increase in vasodilation, detailed further below [48]. Collectively, these responses are described as an exercise mimetic, producing physiological changes similar to those of a moderate-to-vigorous-intensity workout [47] and have led to the implementation of sauna bathing as an alternative therapy to exercise for individuals too impaired to regularly move [42].

Mechanistically, a single session of sauna bathing challenges several enzymes, proteins, and pathways which synergistically work to improve the overall cardiometabolic health. A common observation among individuals suffering from CMD are damaged or dysfunctional proteins [51], as changes in protein structure affect protein function. Heat exposure upregulates heat shock proteins (HSP; ~50%) [45,52] that maintain the cellular environment by protecting against protein denaturation and excessive OS. Specifically, during periods of prolonged thermal stress, HSP serve an important role in preventing damage to proteins by identifying misfolded proteins and preserving their function by correcting their structure [53]. HSP also prevent improper protein folding and mitigate premature apoptosis, thereby promoting the maintenance of cell survival.

Acute exposure to heat also triggers the release of interleukin-6 (IL-6), a pro-inflammatory cytokine [54]. IL-6 plays an important role in the initial stages of inflammation, as its synthesis and release from the liver triggers the liver to produce several acute-phase proteins. Provided that sauna bathing serves as a mild hormetic stressor to physiological systems, it is not surprising to observe an increase in circulating IL-6 levels. However, IL-6 is often overlooked as also an anti-inflammatory cytokine, as IL-6 also dampens the inflammatory cascade by signaling for the release of various anti-inflammatory cytokines such as interleukin-10 (IL-10) [52]. IL-10 is an important cytokine in the healing process, as it ameliorates the inflammation cascade [55] and inhibits several pro-inflammatory cytokines such as tumor necrosis factor-alpha [56] and C-reactive protein (CRP) [57]. CRP is considered one of the strongest independent markers of systemic inflammation [58], and an elevation in CRP plasma levels correlates strongly with the risk of future adverse cardiovascular events [59]. Early in vitro data demonstrated an inverse relationship between IL-10 and CRP levels [60]; in addition, an inverse, dose–response relationship was found between sauna bathing and CRP levels [57]. While only a speculation, as the authors are unaware of any studies directly linking sauna bathing to both IL-10 and CRP levels, it is likely that the lower CRP levels observed in regular sauna users is a result of the chronic synthesis and release of both IL-6 and IL-10. Although the mechanisms linking sauna bathing to the modulation of inflammatory markers in human trials have not yet been elucidated, these findings provide strong implications for the role of sauna bathing in mitigating inflammation in HSO. It is important, however, that future studies aim to investigate the causative factors in sauna bathing that reduce CRP levels and that research extends beyond correlation data. Understanding an exact mechanism for lower CRP levels in regular sauna users would be beneficial in the clinical setting and potentially allow practitioners to recommend sauna bathing as a first approach to treating CMD.

## 4. Cardiometabolic Benefits of Chronic Heat Exposure

Although brief exposure to sauna bathing can result in short-lasting (<1 h) benefits such as reduction of blood pressure and improvement of arterial stiffness, chronic (≥3 weeks) and repeated heat exposure can upregulate several beneficial enzymes and pathways, resulting in greater stress tolerance, a more robust cellular environment, and increased health (Figure 1). Earlier studies showed that chronic sauna bathing improved cellular heat stress tolerance [52], which is linked to a reduction in pro-inflammatory markers [57] and improved insulin sensitivity [32] and has also been shown to improve physical performance [61]. Collectively, these improvements associated with repeated exposure to heat stress are likely attributed to a biological process known as hormesis. Hormesis is defined as a compensatory defense response when repeatedly exposed to low doses of stressors which trigger a plethora of cellular protective mechanisms and protect the cell from subsequent harmful stressors [62]. Therefore, heat stress is a known modulator for the hormetic processes and results in several beneficial cardiometabolic adaptations described below [44].

### 4.1. Cardiovascular Adaptations

Systemically, the cardiometabolic benefits of chronic heat exposure are most pronounced when examining the cardiac function. Laukkanen et al. reported from a large European prospective study that men who attended a sauna bathing session two–three times per week reduced their cardiovascular mortality rates by ~30% and men who attended a sauna session four or more times per week reduced their risk by ~50% [63]. Similarly, Zaccardi et al. observed that men who attended a sauna bathing session two–three times per week reduced their risk for hypertension by ~25% and men who attended a sauna session four or more times per week reduced their risk by ~45% (Figure 2A; [64]). It is now accepted that endothelial dysfunction precedes the development of vascular diseases such as the formation of atherosclerotic plaques along the arterial wall [65]. The improvements observed by Laukkanen et al. [63] and Zaccardi et al. [64] are likely attributed to reductions in endothelium stiffness as a result of elevated nitric oxide synthase levels which increase vessel compliance to dilation and constriction factors, as well as capillary diffusion capability via angiogenesis [66]. In addition to observational studies, the benefits of heat exposure have been described also in other settings, as three weeks of sauna bathing demonstrated improvements in myocardial perfusion in patients suffering from occluded coronary artery-related ischemia [66]. These improvements were collectively the result of increased collateral flow from a reduction in left ventricular end-diastolic pressure and/or improved endothelial function [25]. Although sauna bathing acts as an acute stressor, it might actually mitigate the effects of chronic stress exposure by improving a clinical marker of systemic stress, i.e., heart rate variability [67]. Briefly, heart rate variability represents the fluctuation length between heart beats. Less variance between heart beat intervals represents greater sympathetic activity, a decrease in vagal tone, and a reduced tolerance for the individual to handle or respond to stress. A recent study examined sauna bathing’s effect on heart rate variability and found an improvement of the autonomic system via an increase in vagal tone and a decrease in sympathetic tone and concluded that this could reduce future adverse cardiovascular events [68]. Heat exposure also increases the left ventricle size and subsequently vascular shear stress to meet the demands of blood flow to the skin surface, which is commonly observed during exercise and is a critical trigger of changes in vascular function [69,70]. An increase in left ventricle size through sauna bathing offers clinicians a novel approach for treating heart disease, as left ventricle dysfunction is recognized as a clinical marker of congestive heart failure [71]. The left ventricle is an integral part of the cardiovascular system, whose function is essential for providing sufficient cardiac output to maintain appropriate blood flow to all other organs. Improving the left ventricle function would therefore improve cardiovascular health, as a higher preload from an increase in left ventricle size and a decrease in afterload due to an improvement in endothelium stiffness would result in a lower heart rate and overall lower cardiovascular strain. Studies have confirmed these findings, as data have found that acute (1 day) and chronic (4 weeks) sauna bathing sessions improved left ventricle function [72]. These data are further strengthened by Kihara et al. who observed a significant reduction of systemic vascular resistance and systolic blood pressure (~10 mm Hg, Figure 2B; [73,74]) in subjects attending daily sauna sessions (~15 min) for two weeks.

### 4.2. Metabolic Adaptations

Acute and chronic heat exposure has also been found to improve metabolic functions in the body. One method by which this might occur is through an increase in the expression of 5′-AMP-activated protein kinase (AMPK) [43]. AMPK acts as a cellular energy sensor capable of detecting shifts in the AMP/ATP ratio [75]. When ATP levels decrease, AMPK activity increases, which subsequently upregulates metabolic pathways, such as glycogenolysis and lipolysis, to restore energy availability to cells [76]. The role of AMPK extends beyond merely energy homeostasis, and AMPK has received a lot of attention as a potential target for treating CMD [76,77]. For example, physicians often prescribe metformin as a drug for alleviating symptoms of type 2 diabetes. Zhou et al. showed that metformin primarily works through increasing AMPK expression [78]. Mechanistically, the phosphorylation of AMPK increases lipolytic enzymes, such as hormone-sensitive lipase [79], and glycolytic proteins like, GLUT-4 translocase [80], which together can ease insulin resistance. Interestingly, a single heating session (~2 h) can increase the phosphorylation of AMPK in vitro and in human skeletal muscle [43,81]. This is likely due to sauna bathing acting as an exercise mimetic, as described earlier, which increases caloric expenditure as well as related ventilatory and cardiovascular processes when the body is exposed to heat. The reader should be made aware, however, that although heat stress can increase the expression of AMPK, to date, no longitudinal study has demonstrated a reversal of insulin resistance with chronic sauna bathing, and the reported cascade of events represent merely a speculative explanation for the health benefits described by observational studies.

Further, AMPK has demonstrated an impressive ability to modulate circulating glucose by improving insulin sensitivity and to optimize lipid levels through the phosphorylation of acetyl-CoA carboxylase, which has been proposed as a master regulator of lipid synthesis and non-alcoholic fatty acid liver disease [82]. Dyslipidemia is a strong predictor of the development of CMD and is characterized by a common group of markers (i.e., high-density cholesterol, low-density cholesterol, and triacylglycerols). Regular sauna bathing is known to improve lipid levels, and one study showed that physically active males exposed to 10 sauna bathing sessions experienced a significant decrease in both triacylglycerol and low-density cholesterol levels [32]. A similar study in young, apparently healthy women demonstrated that seven sessions of sauna bathing significantly increased high-density cholesterol levels, although low-density cholesterol and triacylglycerol levels did not statistically differ pre- and post-sauna bathing [83]. The reader should interpret these data with caution however, as a statistical difference in lipid markers may not imply a meaningful change, since the participants of both studies were already deemed healthy. Therefore, the implications of a sauna intervention for apparently healthy individuals are currently limited, and additional studies are warranted. However, regarding HSO associated with insulin resistance, abnormal lipid profiles, and obesity, interventions such as sauna bathing might serve as a potential buffer to these diseased states and improve metabolic health.

The benefits of heat exposure are extensive, as repeated heat exposure is known to stimulate enzymes, pathways, and cellular adaptations that benefit the whole organism. Specifically, repeated heat exposure upregulates HSP and AMPK in cultured cells, which are both known to mediate the expression of a key regulator of mitochondrial biogenesis, i.e., transcriptional coactivator peroxisome proliferator-activated receptor gamma, coactivator-1 alpha (PGC-1α) [28]. These findings have also been confirmed in biopsied skeletal muscle as Hafen et al. demonstrated that a single bout of heat exposure significantly increased the expression of PGC-1α (*p* = 0.04; ~10%) and AMPK (*p* = 0.03; ~33%) [43]. Should these results also occur in vivo, these data would then have far-reaching implications in mitigating the development of CMD or alleviating symptoms such as chronic low-grade inflammation. Although great attention has been directed to the mitochondria for their role in ATP production via oxidative phosphorylation, less discussed is the role that functional mitochondria serve in promoting health through redox homeostasis and mitigation of OS and inflammation [84]. Briefly, the mitochondria serve as a primary source in the defense against OS induced by the generation of excessive free radicals [85]. This is partially accomplished by increasing the activity of mitochondrial transcription factors such as forkhead box, class O (FOXO) and antioxidant response element (ARE), which is mediated by the master regulator of antioxidant and cytoprotective systems, nuclear factor erythroid-2-related factor (Nrf2) [86,87,88]. While PGC-1α regulates the density of the mitochondria within a cell, Nrf2 increases the expression of FOXO and ARE, which both work to improve the activation of several different antioxidant genes and the subsequent detoxification and elimination of free radicals [88,89]. Earlier findings demonstrated that both acute and chronic heat stress can effectively upregulate these aforementioned transcription factors and are able to increase mitochondrial density and function in as little as one week [81,90]. These mitochondria dynamics have important implications for future studies, as mitochondria have demonstrated a continual remodeling process through fission and fusion which serves to maintain and improve their integrity [91]. Mitochondria are often ignored as a potential source of the many health benefits associated with regular sauna bathing. Most sauna bathing data are reported in relation to global health, such as improvements in body composition, inflammation, blood pressure, lipid levels, and markers of heart disease. As research efforts continue to examine sauna bathing as a potential treatment for CMD, it is important to understand the underlying cellular mechanisms, as each metabolic improvement is likely the result of healthier and stronger mitochondria. The synergistic relationship between the mentioned transcription factors and their related signaling pathways results in a more robust cellular and mitochondrial environment which better protects the organism from future stressors, preparing it to react to them appropriately.

## 5. Practical Implications

Military personnel, first responders, police, and firefighters have an increased risk for developing CMD due to the physiological and psychological stress linked to their respective occupations. Over time, chronic exposure to these stressors exacerbates underlying risk factors such as OS and inflammation and can ultimately result in early death. Although it would be fallacious to assume that a sauna intervention can replace the benefits offered by a sound dietary and exercise strategy, sauna bathing is a practical tool that interested individuals can implement to alleviate or mitigate CMD risk factors. For example, one–two sauna bathing sessions per week have been found to significantly decrease sudden cardiac death, coronary heart disease, and overall mortality rates [63]. In a dose–response manner, additional (≥3) sauna bathing sessions can provide further protection to the cardiovascular system such as improved mitochondrial function, reduced inflammation, and an improved lipid profile. Additionally, three–seven sauna bathing sessions per week have demonstrated a ~50% reduction in cardiovascular disease development, risk of stroke, and risk of hypertension [63].

For individuals interested in implementing sauna bathing into their daily routine, observational data suggest sessions initially lasting at least 10 min that should be prolonged to 15 min two–three times a week, to induce the process of acclimation and begin seeing benefits such as a reduced risk of sudden cardiac death, lowered blood pressure and resting heart rate, and an improvement of arterial compliance [63,64]. Following six–seven sauna bathing sessions, the duration of each session can increase in increments of 5 min every 2–3 days, until a timeframe of 45 min is reached, as research shows no meaningful health benefits beyond this duration [31]. Studies incorporating dry saunas have used a range of temperatures (70–95 °C) to induce adaptations; however, in general, the American College of Sports Medicine recommends a temperature range between 70 and 77 °C to achieve the cardiometabolic benefits of sauna bathing [92]. It is assumed that temperatures less than 70 °C may not be sufficient to induce a hormetic effect, while temperatures greater than 100 °C would likely cause cellular damage and premature protein denaturation.

Some concerns should be addressed regarding regular sauna use by men and women. Briefly, in pregnant women, heat exposure has been found to induce some birth defects in the fetus, such as spina bifida [93]. Although observational data have found a negligible association between sauna use and birth defects [49], avoiding sauna use during pregnancy is currently considered best practice, and pregnant women interested in sauna use should consult with their primary physician. Moreover, men experiencing infertility are often recommended by their primary physicians to avoid tight-fitting underwear which places the testicles in close contact with the leg, increasing the temperature of the scrotum and negatively affecting spermatogenesis [94]. In a similar manner, regular exposure to heat, such as with regular sauna use, has been found to significantly reduce sperm motility and sperm count [95]. Garolla et al. showed that two sauna sessions per week (15 min each) for three months reduced sperm mitochondrial function and chromatin condensation in 10 male subjects [95]. However, following six months of terminating sauna use, these findings were completely reversed, and normal baseline values were recorded. Finally, due to the high consumption of alcohol often reported for individuals with HSO [96,97,98], which is usually associated with post-traumatic stress disorder or depression, and the increased risk of heart disease, it is reasonable to speculate that a high percentage of these individuals are currently taking some form of medication. Although regular sauna use is known to improve the markers of heart disease as well as reduce symptoms of depression [99,100], both medication use and alcohol consumption can have severe detrimental and potentially fatal effects if combined concurrently with sauna use. Therefore, alcohol consumption should always be avoided when using a sauna, and any individual taking medication should consult with their primary physician before incorporating sauna bathing into their daily routine. 

## 6. Conclusions

In conclusion, sauna bathing is generally recognized as safe and has emerged as a practical intervention to improve the overall health of individuals serving in an HSO. The physiological benefits that sauna bathing offers are observable at the cellular level and in the whole organism, and these benefits can already be experienced after a single sauna bathing session.

Although data which demonstrates a clear link between heat stress and beneficial cellular mechanisms are lacking, the potential of sauna bathing to mitigate metabolic risk factors is of clinical importance and offers clinicians a low-risk, robust, and innovative treatment approach for combatting CMD often observed in individuals with HSO. Future research should focus on the use of sauna bathing as an intervention for these individuals and examine the effects of both acute and chronic exposure to saunas on physiological responses and, possibly, performance measures.

## Figures and Tables

**Figure 1 ijerph-18-01105-f001:**
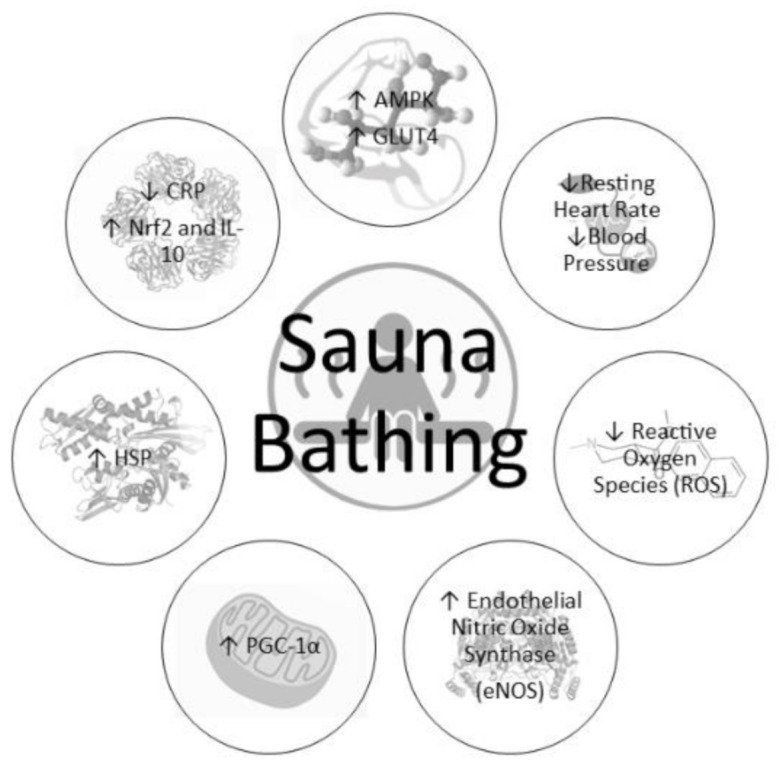
Physiological responses to chronic sauna bathing.

**Figure 2 ijerph-18-01105-f002:**
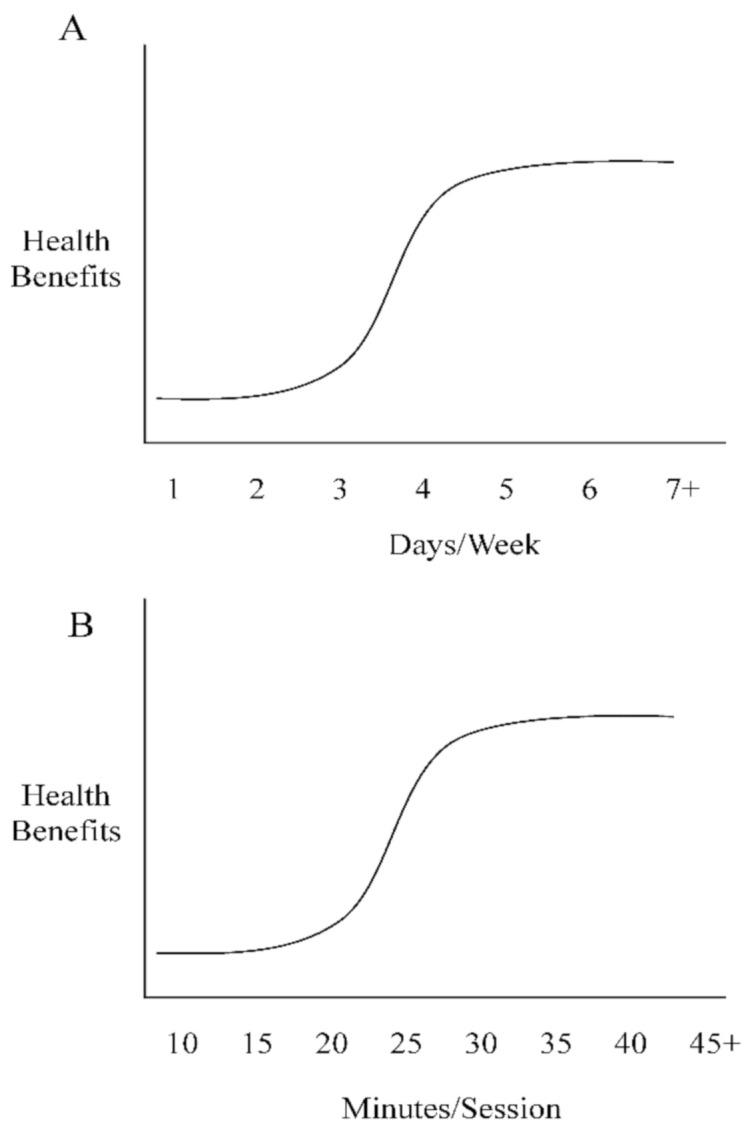
(**A**): Frequency dose-dependent responses to sauna bathing; (**B**): time dose-dependent responses to sauna bathing.

## Data Availability

Not applicable.

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
