# Peer review of "The Cardiometabolic Health Benefits of Sauna Exposure in Individuals with High-Stress Occupations. A Mechanistic Review"

_ijerph, 2021, doi:10.3390/ijerph18031105_

Round 1
Reviewer 1 Report
Review
Overview
The authors conducted this study with the objective of investigating cellular to whole-body mechanisms that dry sauna bathing might offer for improving the cardiometabolic profile. The idea seems to be interesting for a review study. However, I have some questions about topics lacking or that can be better explored. My recommendation is major review.
General comments
Please, add a paragraph about possible benefits of sauna bath on stress markers. For both, acute and chronic sections. Since the main population is high-stress occupation workers, it would be useful to know the effect of sauna bath on the main cause.
Considering physiological responses during sauna bath, what are possible risk for clinical populations (e.g. hypertensives, diabetics)? If there is any risk, what are the strategies to minimize the risk?
Page 7, line 263 - The authors’ say that sauna bathing cannot replace an adequate dietary and regular exercises. Most of the studies manipulated only sauna bathing as an independent variable. Thus, it seems that is still missing data comparing sauna bathing stand- alone with exercise stand-alone and a group with sauna bathing + exercise to investigate if the benefits can be summed, mainly in sedentary or insufficiently trained subjects
Specific comments:
What was the method to search references? What were the database? If there is no method, the authors need to make clear that is a narrative review based on authors’ point of view.
Practice of sauna baths are more common in countries where cold seasons (i.e. low temperatures) are longer. Are the results the same when studies are conducted in tropical countries or comparing summer vs. winter?
Is there any study that investigated the effect of sauna bath on mechanisms of glycemic control? Mainly in diabetics? It seems that all possible benefits on glycemic control is based on AMPK changes.
Page 5, line 190 – An increase around 12 L/min in cardiac output is during sauna bathing, the way is described, it seems a chronic adaptation assessed at rest. “Cheng, J.L.; MacDonald, M.J. Effect of heat stress on vascular outcomes in humans. J Appl Physiol 2019, 126(3), 771-781”
Reviewer 2 Report
The article is basically a literature review and as such is very comprehensive. It makes an attempt to summarize and draw conclusion on the findings and benefits of exposure to dry sauna for HSO such as police and firefighters, etc. It would have been nicer to include more findings other than 2 or 3 graphs.
Reviewer 3 Report
The authors present a nice mechanistic review focused on the acute and chronic effects of exposure to heat, via sauna baths, which seems to be particularly beneficial for subjects who are exposed to systematic stress (high-stress occupations). Indeed, the dry sauna is suggested as a strategy to mitigate the risk of mortality, particularly in this population that is more vulnerable to developing cardiometabolic diseases, being a potential alternative to nutritional and / or exercise interventions, which generally have low adherence.
The paper is very well written, very balanced (volume of text per section), and has sufficient depth and clarity. I particularly liked the section on metabolic adaptations ”, given that it sets out paths for future research. Perhaps the authors may include more proposals for future studies in other sections?!
Although this is not a systematic review, it would be interesting for the authors to clarify the reader (for example at the end of the introduction) about the interest that the scientific community has shown about the theme, and why most studies are carried out using the dry sauna and not other variants.
I also suggested to prepare a summary figure on the physiological effects of exposure to heat. This figure would be very useful (even for future citations), since it will give the manuscript a more didactic character, as intended in this type of review. Additionally, it would be important to clarify what is meant by an acute/chronic exposure to heat in the first sections of the manuscript.
Figures 2A, B raise some doubts; although the idea is to show a dose-response effect, they seem to show an “all or nothing” phenomenon, which does not seem to be exactly what is recognized in the literature. I suggest eliminating these figures or at least reviewing them for their suitability. Alternatively, consider elaborating a summary table in the “practical implications” section.
Many congratulations to the authors for the relevance of this review, which was carried out with depth, clarity, organization and supported by a careful and current set of specialized literature.
Round 2
Reviewer 1 Report
Dear authors
Most of my questions were properly addressed but one the specific comment below considering glycemic control. Since the singularity of experiments and populations, the authors should clarify when the results were found in healthy subjects. Besides, they need tone down with conclusions for metabolic functions due to the few literature considering each outcome [glycemic control (AMPK), lipids, PGC-1, inflammatory markers]
My recommendation is minor review.
Author Response
Please see attachment

This manuscript is a resubmission of an earlier submission. The following is a list of the peer review reports and author responses from that submission.